# Using Sentinel-2-Based Metrics to Characterize the Spatial Heterogeneity of FLEX Sun-Induced Chlorophyll Fluorescence on Sub-Pixel Scale

Nela Jantol [1,*,†], Egor Prikaziuk [2,†], Marco Celesti [3], Itza Hernandez-Sequeira [4], Enrico Tomelleri [5], Javier Pacheco-Labrador [6], Shari Van Wittenberghe [7], Filiberto Pla [4], Subhajit Bandopadhyay [8], Gerbrand Koren [9], Bastian Siegmann [10], Tarzan Legović [1,11,12], Hrvoje Kutnjak [13] and M. Pilar Cendrero-Mateo [7]

1. Laboratory for Remote Sensing and GIS, Oikon Ltd.-Institute of Applied Ecology, 10020 Zagreb, Croatia
2. Faculty Geo-Information Science and Earth Observation (ITC), University of Twente, 7541 AE Enschede, The Netherlands
3. HE Space for ESA—European Space Agency, European Space Research and Technology Centre (ESA-ESTEC), Keplerlaan 1, 2201 AZ Noordwijk, The Netherlands
4. Institute of New Imaging Technologies, University Jaume I, 12071 Castellón de la Plana, Spain
5. Faculty of Sciene and Technology, Free University of Bozen-Bolzano, 39100 Bolzano, Italy
6. Max Planck Institute for Biogeochemistry, 07745 Jena, Germany
7. Laboratory of Earth Observation, Image Processing Laboratory, University of Valencia, C/Catedrático Agustin Escardino 9, 46980 Paterna, Spain
8. Department of Geography and Environmental Science, University of Southampton, Southampton SO17 1BJ, UK
9. Copernicus Institute of Sustainable Development, Utrecht University, 3508 TC Utrecht, The Netherlands
10. Institute of Bio- and Geosciences, Plant Sciences (IBG-2), Forschungszentrum Jülich, 52428 Jülich, Germany
11. Division for Marine and Environmental Research, Ruđer Bošković Institute, Bijenička c. 54, 10000 Zagreb, Croatia
12. LIBERTAS International University, 10000 Zagreb, Croatia
13. Division of Plant Science, Faculty of Agriculture, University of Zagreb, Svetošimunska 25, 10000 Zagreb, Croatia
* Correspondence: njantol@oikon.hr
† These authors contributed equally to this work.

**Abstract:** Current and upcoming Sun-Induced chlorophyll Fluorescence (SIF) satellite products (e.g., GOME, TROPOMI, OCO, FLEX) have medium-to-coarse spatial resolutions (i.e., 0.3–80 km) and integrate radiances from different sources into a single ground surface unit (i.e., pixel). However, intrapixel heterogeneity, i.e., different soil and vegetation fractional cover and/or different chlorophyll content or vegetation structure in a fluorescence pixel, increases the challenge in retrieving and quantifying SIF. High spatial resolution Sentinel-2 (S2) data (20 m) can be used to better characterize the intrapixel heterogeneity of SIF and potentially extend the application of satellite-derived SIF to heterogeneous areas. In the context of the COST Action Optical synergies for spatiotemporal SENsing of Scalable ECOphysiological traits (SENSECO), in which this study was conducted, we proposed direct (i.e., spatial heterogeneity coefficient, standard deviation, normalized entropy, ensemble decision trees) and patch mosaic (i.e., local Moran's I) approaches to characterize the spatial heterogeneity of SIF collected at 760 and 687 nm ($SIF_{760}$ and $SIF_{687}$, respectively) and to correlate it with the spatial heterogeneity of selected S2 derivatives. We used HyPlant airborne imagery acquired over an agricultural area in Braccagni (Italy) to emulate S2-like top-of-the-canopy reflectance and SIF imagery at different spatial resolutions (i.e., 300, 20, and 5 m). The ensemble decision trees method characterized FLEX intrapixel heterogeneity best ($R^2 > 0.9$ for all predictors with respect to $SIF_{760}$ and $SIF_{687}$). Nevertheless, the standard deviation and spatial heterogeneity coefficient using k-means clustering scene classification also provided acceptable results. In particular, the near-infrared reflectance of terrestrial vegetation (NIRv) index accounted for most of the spatial heterogeneity of $SIF_{760}$ in all applied methods ($R^2 = 0.76$ with the standard deviation method; $R^2 = 0.63$ with the spatial heterogeneity coefficient method using a scene classification map with 15 classes). The models

developed for $SIF_{687}$ did not perform as well as those for $SIF_{760}$, possibly due to the uncertainties in fluorescence retrieval at 687 nm and the low signal-to-noise ratio in the red spectral region. Our study shows the potential of the proposed methods to be implemented as part of the FLEX ground segment processing chain to quantify the intrapixel heterogeneity of a FLEX pixel and/or as a quality flag to determine the reliability of the retrieved fluorescence.

**Keywords:** spatial heterogeneity; vegetation indices; biophysical traits; SIF; hyperspectral sensor; Sentinel-2; FLEX; Braccagni

## 1. Introduction

The dynamic nature of Sun-Induced chlorophyll Fluorescence (SIF) makes it highly recommended to characterize its spatiotemporal heterogeneity before using it to monitor vegetation from space [1]. SIF is the light emitted by plants within the spectral window of 650–800 nm and is characterized by a peak in the red (685 m) and far-red (740 nm) regions of the spectrum. During the last years, satellite missions with coarse-to-moderate spatial resolution (e.g., GOME-2 40 × 80 km, GOSAT 10 km diameter, OCO-2 1.29 × 2.25 km, OCO-3 1.6 × 2.2 km, TROPOMI 3.5 × 7.5 km–3.5 × 5.5 km since August 2019) have been used to produce global maps of SIF [2–4]. Furthermore, the European Space Agency (ESA) is planning to launch the FLuorescence EXplorer (FLEX) satellite mission in 2025, with an improved spatial resolution of 300 × 300 m, being the first SIF-dedicated satellite mission [5,6].

SIF measured from satellites can provide relevant information about the actual plant photosynthetic capacity, linking the leaf-level molecular mechanism to Earth-system science [7]. SIF has been used in different studies to improve remote estimations of Gross Primary Production (GPP) [8–11], for early stress detection [12–18], and to study vegetation dynamics in different climate zones (e.g., [19–21]).

However, SIF is a highly dynamic signal, and intrapixel spatiotemporal variations of illumination conditions, vegetation fractional cover, or land use can mislead the quantification and consequently the interpretation of satellite SIF estimates [22,23].These include inaccurate retrieval of SIF when intrapixel heterogeneity includes differences in chlorophyll content and/or vegetation fractional cover that alter the measured reflected radiance spectral shape, as well as inaccurate quantification of SIF emitted by the photosynthetic surface when vegetation structure and self-shading within the canopy are not considered [7]. For example, Cogliati et al. [24] showed a decrease in the SIF retrieval performance at low leaf area index. Kováč et al. [25] observed that the dynamics of canopy shadow fraction in forest, in addition to changes induced by sunlight, influence the daily variability of SIF. Zarco-Tejada et al. [26] developed the FluorFLIM model to simulate SIF in heterogeneous canopies at coarse spatial resolution (50 m). They showed that SIF retrieved from mixed pixels (containing pure canopy, shade, and soil) resulted in a weak correlation ($R^2 = 0.38$) with stomatal conductance compared to ground measurements, due to the effects of different vegetation fractional cover. Moreover, Tagliabue et al. [27] showed that in mixed forests, spatial heterogeneity plays a crucial role in controlling the relationship between far-red SIF and GPP. Moncholi-Estornell et al. [28] showed that normalizing the SIF signal emitted from the top of the canopy by the fractional cover of sunlit vegetation improves the estimation of the effective fluorescence flux, reducing the error from 36% to 18% (red fluorescence) and from 24% to 6% (far-red fluorescence), respectively. These studies show how an inaccurate characterization of the spatial heterogeneity of a fluorescence pixel can lead to errors in the estimation of the fluorescence signal. Therefore, in this paper we will focus on characterizing the spatial heterogeneity of a SIF pixel.

Spatial heterogeneity has been defined differently in multiple disciplines (e.g., ecology, geography, landscape, Remote Sens., etc.). In this article, we define spatial heterogeneity as the complexity and variability of a system property in space. Complexity refers

to qualitative variables (i.e., land cover), while variability refers to quantitative variables (leaf area index (LAI), fractional cover (fCover), chlorophyll content (Chl), etc.) [29]. In both cases, the spatial distribution of these variables within an area can be described using frequency (spatial variability) and patterns (spatial structures) [30,31]. Following the above definitions, spatial heterogeneity has been quantified from remotely sensed imagery using two basic approaches: (a) the direct approach, where reflectance, reflectance indices and/or retrieved biophysical variables were used to quantify spatial heterogeneity, and (b) the patch mosaic approach, where the image was classified into homogeneous mapping units [32–34].

With respect to fluorescence heterogeneity, not many studies have been conducted. For example, Rossini et al. [1] evaluated an optimal sampling strategy to characterize the spatial representativeness of SIF using the Normalized Vegetation Index (NDVI). Buman et al. [35] analyzed radiometric, spectral, and spatial uncertainties that affect the accuracy of SIF retrievals using HyPlant and FloX spectrometers. Although these studies focused on guiding cal/val activities for the FLEX mission, they did not evaluate the contribution of different vegetation indices (i.e., near-infrared reflectance of terrestrial vegetation, chlorophyll red-edge) and biophysical traits (i.e., leaf area index, vegetation fractional cover) and/or compare different methods for characterizing SIF spatial variability.

The main challenge in characterizing the spatial heterogeneity of SIF is the fact that SIF cannot be confidently predicted from reflectance-based information. The lack of high spectral and spatial resolution satellite data required to properly retrieve the functional-based (i.e., APAR and NPQ) traits driving the SIF dynamic changes is also a challenge. However, what we do have available are complementary data like the high spatial resolution Sentinel-2 (S2, $20 \times 20$ m) based products (i.e., reflectance data, vegetation indices, and biophysical traits), which we hypothesize can be used to characterize the spatial heterogeneity of FLEX Sun-Induced chlorophyll Fluorescence on a sub-pixel scale. Within this context, and in the frame of the SENSECO COST Action, in this study we propose to use S2 imagery information to quantify the spatial heterogeneity of a FLEX pixel ($300 \times 300$ m). Since at the moment FLEX maps are not yet available, we have taken advantage of HyPlant airborne sensor high spatial resolution observations to perform this study [36]. HyPlant reflectance images were resampled and convoluted to mimic S2 ($20 \times 20$ m) spatial and spectral resolution. After that, spatial heterogeneity metrics were computed on simulated S2 data within the FLEX pixels, using HyPlant SIF images ($4.5 \times 4.5$ m) as a reference. Later, several spatial heterogeneity characterization methods like direct (spatial heterogeneity coefficient, standard deviation, entropy, ensemble decision trees) and patch mosaic (local Moran's I index) were implemented using both HyPlant S2-based products and SIF images. The objective of this study is to determine which methods and S2-based products are suitable to characterize the spatial heterogeneity of a FLEX pixel.

## 2. Materials and Methods

### 2.1. Study Area

The study took place in a 14 km$^2$ rural area located in Braccagni, central Italy (42.82°N; 11.07°E) (Figure 1). The scene was mostly agricultural and encompassed different summer crops (i.e., tomato, corn, sorghum), which are irrigated during June-September (FlexSense final report—ESA contract no. 4000125402/18/NL/NA). According to the Urban Atlas 2018 land-use map (European Union, Copernicus Land Monitoring Service 2018, European Environment Agency (EEA)), arable lands dominate the site (~82%), followed by pastures (~7%), isolated structures (2.50%), industrial, commercial, private units (~1%), sports and leisure facilities (~1%), and other roads and lands associated with fast roads (~1%). Permanent crops (e.g., vineyards, orchards), water, forests, green urban areas, and discontinuous urban fabric each represent less than 1% of the site.

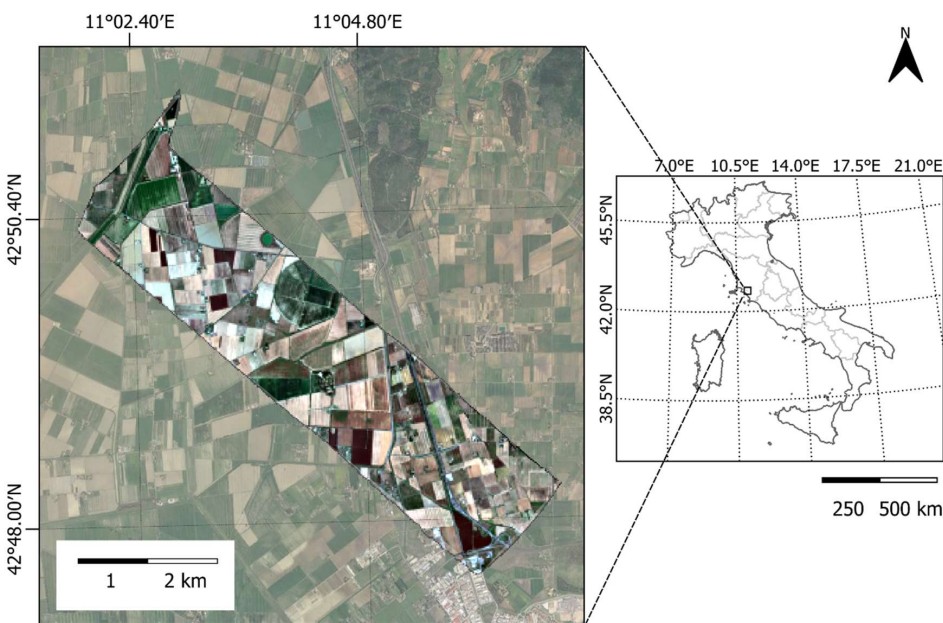

**Figure 1.** Study area, Braccagni, Italy. Map of the area on the left was produced using Sentinel-2 RGB bands (B4-B3-B2).

## 2.2. Airborne Data

HyPlant airborne image data were acquired on 30 July 2018 over Braccagni, Italy, as a part of FlexSense 2018 campaign (ESA Contract No. 4000125402/18/NL/NA). The HyPlant imaging spectrometer consists of two sensor modules, the DUAL and FLUO module, covering spectral ranges from 400 to 2500 nm (spectral sampling interval, SSI = 1.71 nm (400–1000 nm), SSI = 5.58 nm (1000–2500 nm)) and 670 to 780 nm (SSI = 0.11 nm), respectively.

The technical specifications of HyPlant FLUO allows for the retrieval of SIF in the $O_2A$ ($SIF_{760}$) and $O_2B$ ($SIF_{687}$) absorption bands [36,37]. The spectral fitting method (SFM), developed by Cogliati et al. [38] and later adapted to airborne data [39], was used to retrieve SIF at 760 and 687 nm. The recorded HyPlant images cover an area of approximately 14 km$^2$ and were acquired heading in the northern direction from 350 m above ground level at 11:40 local time. The image data of both sensors were processed and georectified according to the HyPlant processing chain presented in Siegman et al. [37].

## 2.3. Data Processing Description and Heterogeneity Methods Evaluation

In this section, we provide an overview of the different steps in our analysis and how they relate to each other. These steps are described briefly here and in more detail in the following sections. The data processing flow of this study is summarized in the following steps (Figure 2).

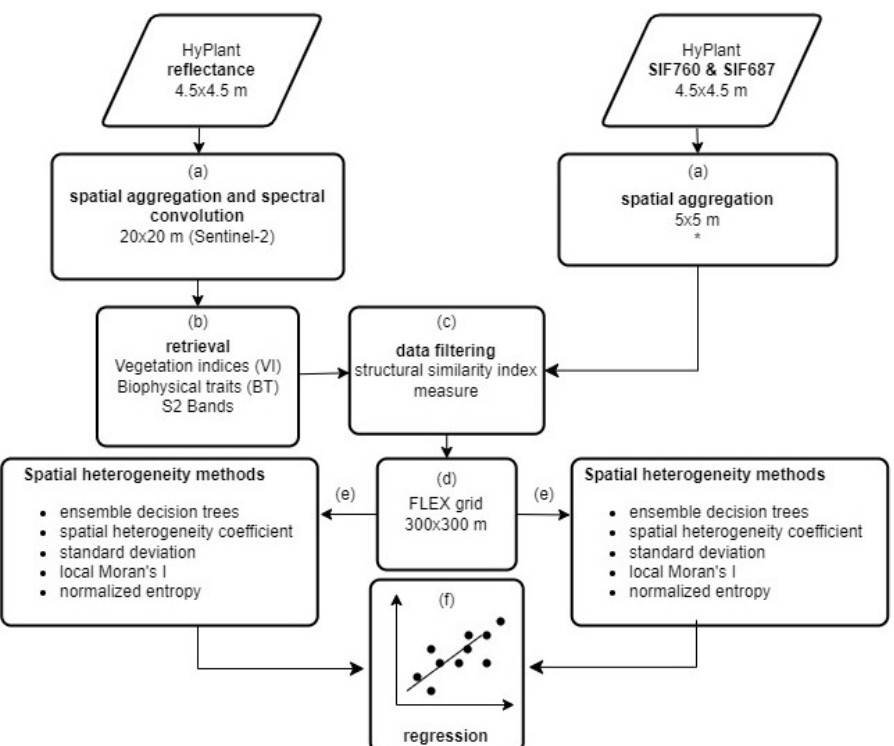

**Figure 2.** Workflow diagram. (**a**) HyPlant reflectance image (4.5 × 4.5 m) was aggregated to mimic S2 spectral and spatial resolution (13 bands and 20 × 20 m ~ S2-R$_{20}$). At the same time, HyPlant fluorescence products were spatially aggregated to 5 × 5 m resolution (SIF$_{687,5}$ and SIF$_{760,5}$) (see data preparation section). * For the ensemble decision trees method, SIF was additionally aggregated to 300 × 300 m. (**b**) Synthetic S2-R$_{20}$ bands were used to obtain the biophysical traits (S2-BT$_{20}$) and vegetation indices (S2-VI$_{20}$), which were later used to characterize the spatial heterogeneity of SIF (Tables 1 and 2). (**c**) The Structural Similarity Index Measure (SSIM) was implemented to filter the input data (i.e., S2 bands, VIs, BT) used in the study. (**d**) To determine the spatial heterogeneity of a FLEX pixel, a 300 × 300 m grid was applied to the S2 synthetic (S2-R$_{20}$, S2-BT$_{20}$ and S2-VI$_{20}$) and SIF (SIF$_{687,5}$ and SIF$_{760,5}$) resampled images. Each FLEX pixel potentially contained 15 × 15 S2 pixels and 60 × 60 SIF 5 × 5 m pixels. (**e**) Different heterogeneity methods (see methods to characterize sun-induced chlorophyll fluorescence heterogeneity section) were applied to the S2 and HyPlant SIF products using the 300 × 300 FLEX grid defined in step (**d**). A FLEX heterogeneity product was obtained for each S2 predictor (S2-R$_{20}$, S2-BT$_{20}$ and S2-VI$_{20}$) and SIF reference data (SIF$_{687,5}$ and SIF$_{760,5}$). (**f**) Finally, we compared S2 vs. SIF heterogeneity products using linear regression (see models' performance section).

### 2.4. Data Preparation

The original spatial resolution of the image data recorded by both HyPlant sensor modules was 4.5 m. The data were then spatially resampled to match the 20 m S2 and 300 m FLEX grids (the Sentinel-3 reference grid was used for FLEX because FLEX will be spatially consistent with Sentinel-3). Spatial resampling and spectral convolution were performed as follows:

-   From the FLUO sensor (4.5 × 4.5 m), SIF in the O$_2$A (SIF$_{760}$) and O$_2$B (SIF$_{687}$) bands was spatially aggregated in the software SAGA ([40], version 2.3.2) using the nearest neighbor algorithm to downscale it from 4.5 m to 5 m (SIF$_{760,5}$ and SIF$_{687,5}$). SIF was not aggregated to 20 m because we used the SIF image data as a reference, and therefore decreasing the resolution to 20 m would result in a loss of information needed for the characterization analysis. We did not exclude negative SIF values inherent to SIF retrieval uncertainty. Although they lack physical meaning (negative SIF is physically

not possible), removing them would arbitrarily bias the resampled data. Therefore, retrieval uncertainty contributes unavoidably to the spatial heterogeneity of SIF.

- Top-of-canopy reflectance data from the DUAL module of HyPlant (626 bands in total) were first spatially aggregated from 4.5 m to 20 m to mimic S2 pixels. We, again, used the software SAGA and information about the S2 grid to perform this task [40] (version 2.3.2). Spatial resampling was performed using the mean (cell area weighted) downscaling method. The output image was then processed in R using the hsdar package [41] for spectral convolution. This resulted in 13 synthetic S2 spectral reflectance bands at 20 m spatial resolution (S2-R$_{20}$), which later were used to retrieve the biophysical traits (S2-BT$_{20}$) and vegetation indices (S2-VI$_{20}$) used to characterize SIF spatial heterogeneity (Tables 1 and 2).

**Table 1.** Summary of the Sentinel-2-based vegetation indices used to determine SIF spatial heterogeneity within the 300 × 300 m resolution FLEX pixels. Sentinel-2 central wavelength: B1 (443 nm), B2 (490 nm), B3 (560 nm), B4 (665 nm), B5 (705 nm), B6 (740 nm), B7 (783 nm), B8 (842 nm), B8a (865 nm), B9 (940 nm), B10 (1375 nm), B11 (1610 nm) and B12 (2190 nm).

| Vegetation Index (VI) | General/Sentinel-2 Formula | Description |
|---|---|---|
| Normalized difference vegetation index (NDVI) | NDVI = (NIR − RED)/(NIR + RED)<br>NDVI = ((B8A − B4)/(B8A + B4)) | Indicator of green vegetation [42]. |
| Near-infrared reflectance of terrestrial vegetation (NIRv) | NIRv = NIR×((NIR − RED)/(NIR + RED))<br>NIRv = B8A×((B8A − B4)/(B8A + B4)) | Proportion of pixel reflectance due to vegetation in the pixel; strongly correlated with SIF [21,43,44]. |
| Chlorophyll red-edge (ChlRE) | ChlRE = ([760:800]/[690:720]) − 1<br>ChlRE = B7/B5-1 | Estimates chlorophyll content in leaves [45]. |
| Enhanced vegetation Index (EVI) | EVI = 2.5×(NIR − RED)/((NIR + 6 × RED − 7.5 × BLUE) + 1)<br>EVI = 2.5×(B8A − B4)/(B8A + 6 × B4 − 7.5 × B2) + 1) | Indicator of green vegetation similar to NDVI, but corrects for some atmospheric conditions and is more sensitive to dense vegetation [46,47]. |
| Moisture content (MSI) | MSI = SWIR/NIR<br>MSI = B11/B08A | Indicator of leaf water content—higher values indicate high water stress with less water content and vice-versa [48,49]. |

**Table 2.** Summary of the Sentinel-2-based biophysical traits used to determine sun-induced fluorescence spatial heterogeneity within FLEX pixels of 300 × 300 m resolution. Biophysical traits were retrieved using the Sentinel-2 ToolBox Biophysical processor [50].

| Biophysical Trait (BT) | Description |
|---|---|
| Fraction of Absorbed Photosynthetically Active Radiation (fAPAR) | Fraction of the down-welling photosynthetically active radiation that is absorbed by the canopy [51]. |
| Leaf Area Index (LAI) | Quantifies the amount of leaf material in a canopy. It is the ratio of one-sided leaf area per unit ground area [52,53]. |
| Fraction of green Vegetation Cover (fCover) | Quantifies the fraction of ground covered by green vegetation [54]. |
| Leaf Chlorophyll Content (LCC) | Leaf chlorophyll content (μg of chlorophyll per cm$^2$ of leaf area) was computed from the retrieved canopy chlorophyll content (CCC), dividing it to the retrieved LAI [55]. |

### 2.5. Predictor Selection

We selected only the S2 predictors that presented spatial patterns similar to those of SIF using the structural similarity index measure (SSIM). SSIM is widely used for digital images and videos to quantify the similarity between two images. It was developed by [56] and uses structural information related to the spatial arrangement of pixels. We chose this approach to compare the spatial similarity of SIF$_{687,20}$ and SIF$_{760,20}$ images (aggregated to 20 m only for predictor selection) with S2 predictors (spectral reflectance bands, vegetation indices and biophysical traits, 20 m resolution). The images were normalized, scaled between 0 and 1 in accordance with min-max values, and SSIM was computed for all 300 × 300 m pixels (FLEX pixels based on Sentinel-3 grid with assigned pixel ID-s). Tukey's test was then used to compare the means of the SSIM values for the predictors (S2-R$_{20}$,

S2-BT$_{20}$, and S2-VI$_{20}$) and SIF (SIF$_{687,20}$, SIF$_{760,20}$). Values around 0 mean that there is no similarity between the SIF and the S2 predictor patterns, while values towards ±1 indicate high similarity. S2 bands B1, B2, B3, B4, B5, B10, B11 and B12 were not similar to SIF$_{687,20}$ and SIF$_{760,20}$ (Figure 3) and were not used for further analysis, meaning that all the visible and SWIR bands were removed, leaving only NIR and red-edge bands. It is worth noting that SIF$_{687,20}$ itself was not so similar to SIF$_{687,20}$, nor to any S2-BT$_{20}$ or S2-VI$_{20}$ product.

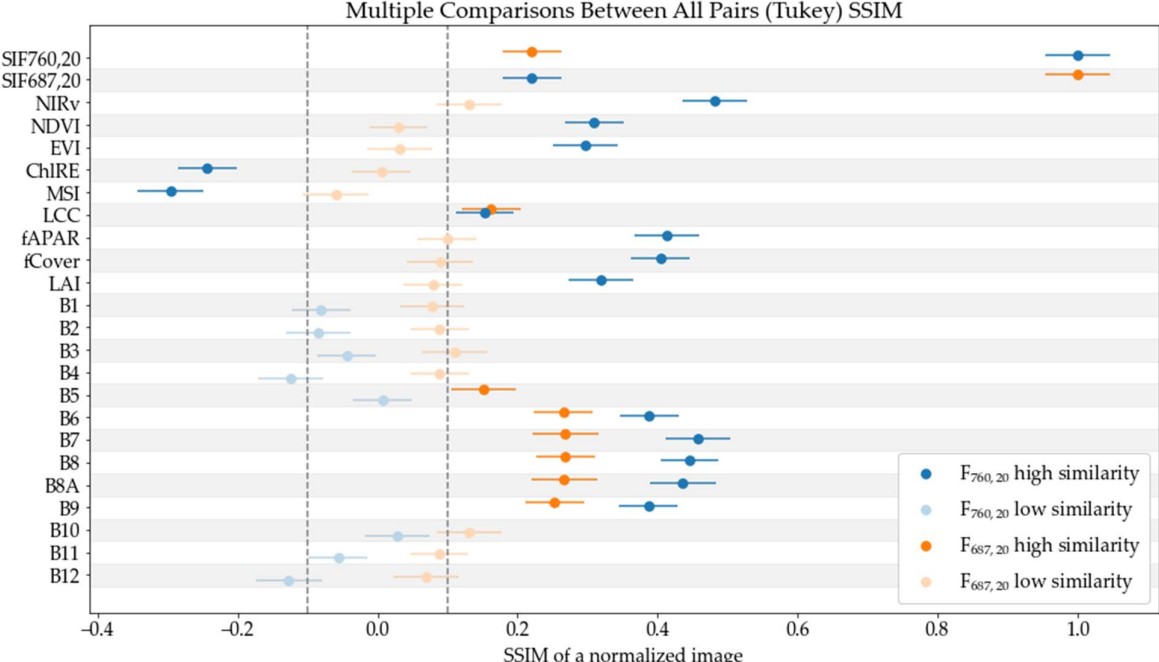

**Figure 3.** Tukey's test applied to the Structural Similarity Index Measure (SSIM) used to measure the similarity between two normalized images (SIF$_{760,20}$, SIF$_{687,20}$, and respective Sentinel-2 predictors). Dashed vertical lines indicate the similarity threshold of ±0.1 SSIM. Bands with SSIM values above this threshold for both SIF$_{760}$ and SIF$_{687}$ were used for further analysis.

### 2.6. Methods to Characterize Sun-Induced Chlorophyll Fluorescence Heterogeneity

This study evaluated the ability of one patch mosaic and four direct methods to characterize the heterogeneity of HyPlant SIF at 5 m and S2 information at 20 m spatial resolution within 300 × 300 m. The methods implemented in this study are described in Table 3. A more detailed explanation of each method is described in Appendix A. In addition, we tested other approaches such as cluster entropy and a fuzzy approach, but these failed to characterize SIF heterogeneity (Appendix B) and were excluded from further analyses. Heterogeneity measures with different methods were expressed as single values per predictor and 300 m pixel, each labeled with a numeric ID. Sub-pixels where fluorescence retrieval failed (NaN values) were omitted in the calculations, meaning that the total number of sub-pixels varied within the FLEX pixels. Additionally, we discarded pixels on the edge of the imagery to avoid large fractions of missing pixels.

We evaluated the potential of each method to characterize SIF heterogeneity by comparing the heterogeneity metrics calculated from the HyPlant SIF and S2 predictors and calculating the square of Pearson's correlation coefficient ($R^2$). Other goodness-of-fit metrics (e.g., coefficient of determination, root-mean-square error, and bias) would not produce meaningful comparisons for this task due to differences in the units of SIF (W m$^{-2}$ sr$^{-1}$ μm$^{-1}$) and predictors (unitless for vegetations indices, fAPAR, fCover; m$^2$ m$^{-2}$ for LAI; sr$^{-1}$ for reflectance bands; μmol m$^{-2}$ for LCC) . Due to the squared nature of the spatial heterogeneity coefficient (variance is a square of the standard deviation, see Table A1) it was unsquared using natural logarithm transformation to avoid heteroscedasticity while applying the linear regression.

**Table 3.** Summary of the different methods used to characterize SIF spatial heterogeneity within the FLEX spatial resolution (300 × 300 m). Predictors reported are single-band reflectance at 20 m (S2-$R_{20}$); spectral vegetation indices at 20 m (S2-$VI_{20}$); biophysical traits at 20 m (S2-$BT_{20}$); or a combination of several of these (S2-$R_{multi20}$/S2-$VI_{multi20}$/S2-$BT_{multi20}$).

| Method Name | Heterogeneity Definition | Predictors | Reference |
|---|---|---|---|
| Local Moran's I | The classification of sub-pixels is based on the spatial autocorrelation metric Moran's I, whose statistical significance is defined by permutations (bootstrap). Heterogeneity is expressed as the fraction of sub-pixels belonging to the "no class" or "single pixel cluster class" over the total number of sub-pixels in a 300 × 300 FLEX pixel. | S2-$VI_{20}$ <br> S2-$BT_{20}$ <br> S2-$R_{20}$ | [57] |
| Spatial heterogeneity coefficient | Interclass and intraclass differences combined with their spatial distribution. Classes are generated using supervised and unsupervised approaches in the form of Scene Classification Maps (SCLs). | S2-$VI_{20}$ <br> S2-$BT_{20}$ <br> S2-$R_{20}$ | [31] |
| Standard deviation | Standard deviation over the total number of sub-pixels in a 300 × 300 FLEX pixel. | S2-$VI_{20}$ <br> S2-$BT_{20}$ <br> S2-$R_{20}$ | / |
| Ensemble decision trees | Four different machine learning algorithms to predict $SIF_{\lambda,20}$ as a function of $SIF_{\lambda,300}$, and S2-$VI_{20}$, S2-$BT_{20}$, S2-$R_{20}$, S2-$R_{multi20}$, S2-$VI_{multi20}$, S2-$BT_{multi20}$: eXtreme Gradient Boosting, Random Forests, Support Vector Machines, and Neural Networks. The most accurate algorithm (Random Forest) was used to upscale SIF from FLEX to S2 spatial resolution. | S2-$VI_{20}$ <br> S2-$BT_{20}$ <br> S2-$R_{20}$ <br> S2-$R_{multi20}$ <br> S2-$VI_{multi20}$ <br> S2-$BT_{multi20}$ | [58] |
| Normalized Entropy | Heterogeneity was quantified using the concept of entropy that measures the average information content. The entropy was normalized by the entropy of the uniform distribution (Emax with N = sub-pixels in a FLEX pixel). | S2-$VI_{20}$ <br> S2-$BT_{20}$ <br> S2-$R_{20}$ | [59] |

### 2.7. Outliers' Distribution

We used the Root Mean Square Error (RMSE) metric to compare the spatial heterogeneity values calculated for the reference (SIF) and model predictor pixels. The top five pixels with the highest RMSE were defined as "outliers", and their structure was investigated in more detail.

## 3. Results

To better understand the results of the different models, the results section is divided into two blocks. First, the spatial distribution of study area land cover types as well as SIF, VIs, and biophysical traits are described. Histograms were made to visualize the variability and skewness of the dataset. Second, the performance of each model was evaluated (SIF vs. predictors), where the pixels with the highest RMSEs (outliers) were examined in more detail. Finally, the heterogeneity maps were described for the best-performing models, highlighting the pixels with the highest and lowest heterogeneity.

### 3.1. Field Site Characterization

Spatial Analysis

The supervised classification of the S2 image of the area resulted in the scene classification map with five land cover classes shown in Figure 4a with the following percentages for the scene: cropland 76%, pasture 5%, forest 5%, water 0.2% and 13% other (unclassi-

fied). Similarly, Figure 4b presents the results of the k-means unsupervised classification with 15 classes.

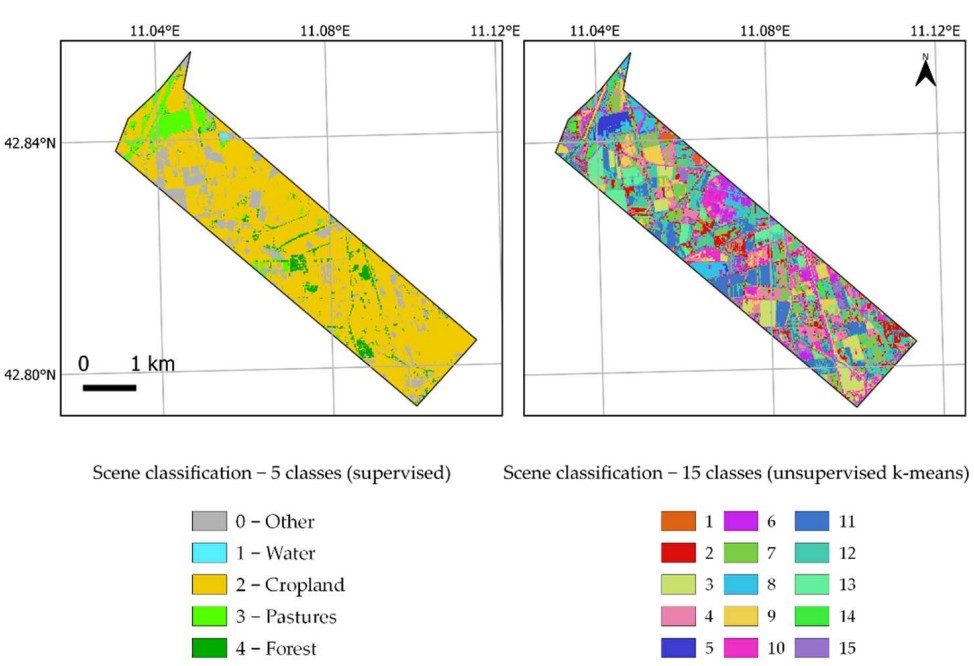

**Figure 4.** Scene classification maps: (**a**) Map produced using supervised classification with 5 classes; (**b**) Map produced using k-means algorithm with 15 classes.

Fluorescence radiances ranged from $-0.15$ to $0.34$ W m$^{-2}$ um$^{-1}$ sr$^{-1}$ for SIF$_{760,5}$ and from $-0.13$ to $0.2$ W m$^{-2}$ um$^{-1}$ sr$^{-1}$ for SIF$_{687}$ (Figure 5a). Higher SIF$_{760,5}$ values were observed in the northern and central parts of the image, which correspond to green pasture and forested areas, respectively (Figure 4a).

SIF$_{687,5}$ had higher values at the southern image boundary that were not observed at the northern boundary (Figure 5a). The patterns of vegetation indices and biophysical trait maps (Figure 5b,c) are consistent with those of SIF, with higher values of NIRv, NDVI, EVI, fAPAR, fCover, LAI, and low values of ChlRE, MSI and LCC in the same areas of the image. For example, the circular shape area in the middle of the image was a 1 km diameter irrigated corn crop and was highlighted in these maps. For MSI, a measure of vegetation water content, higher values indicate lower water content. Finally, the S2 reflectance bands B6, B7, B8, B8A and B9 all followed a mutually similar distribution (Figure 5d).

The frequency distribution of the normalized SIF, Vis, BTs and S2 reflectance bands is shown in Figure 6. Both SIF$_{760,20}$ and SIF$_{687,20}$ showed a normal distribution, with SIF$_{760,20}$ having a higher frequency of lower values than SIF$_{687,20}$ (Figure 6a). When analyzing the distribution of Vis, a similar normal distribution pattern was observed for ChlRE and MSI, whereas the peaks for EVI, NDVI and NIRv (Figure 6b) distributions were skewed towards lower values. Distributions of biophysical traits fAPAR, fCover, and LAI (Figure 6c) also had higher frequency of lower values, but the skew is much more pronounced compared to the Vis. All S2 reflectance band distributions were slightly skewed to the left with a higher frequency of higher values (Figure 6d).

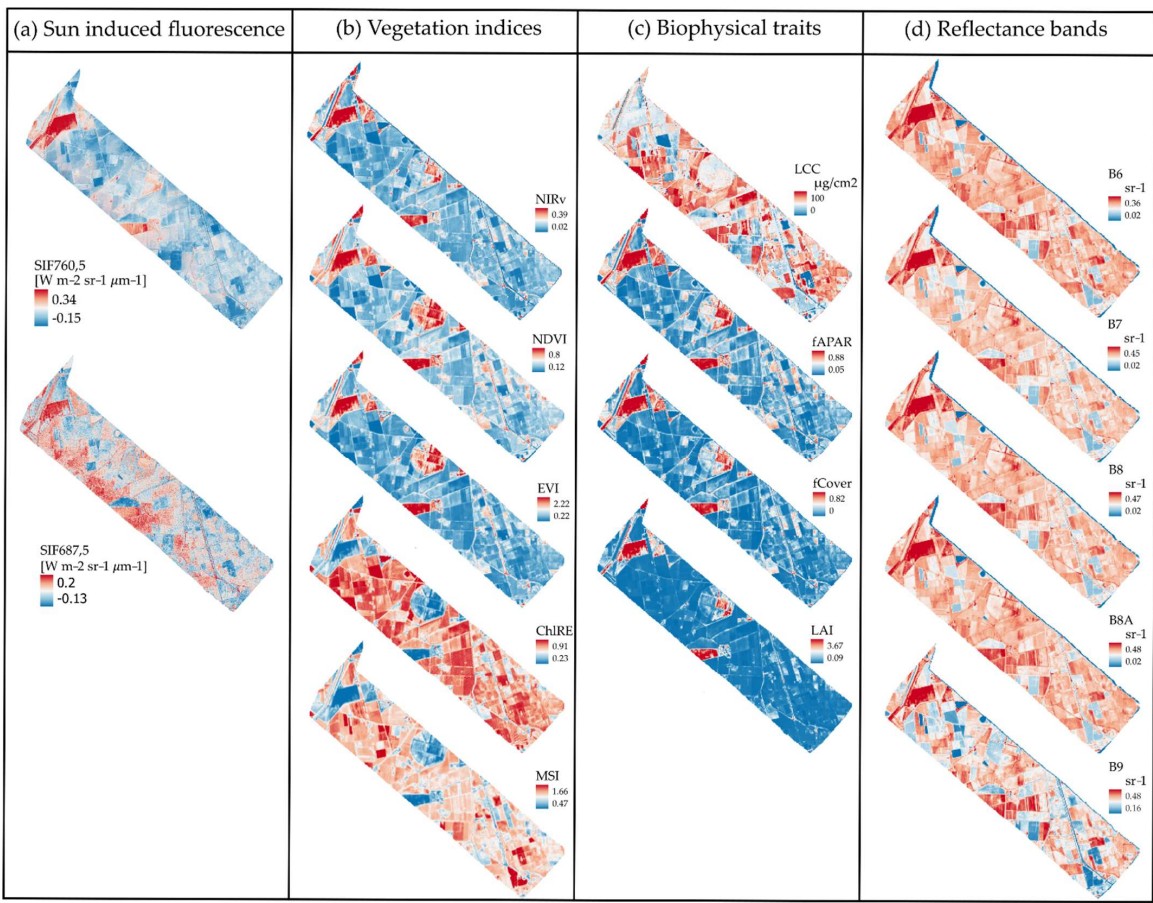

**Figure 5.** Dataset imagery: (**a**) Sun-induced fluorescence with 5 × 5 m resolution; (**b**) Vegetation indices maps from Sentinel-2 data NIRv, NDVI, EVI, ChlRE, MSI with 20 × 20 m resolution; (**c**) Biophysical traits maps for LCC, fAPAR, fCover and LAI at 20 × 20 m resolution; (**d**) Reflectance bands maps from Sentinel-2 data B6, B7, B8, B8A, B9 with 20 × 20 m resolution. Values in maps are shown as 2nd and 98th percentiles of the raster band values. Lower values are shown in blue, higher values in red.

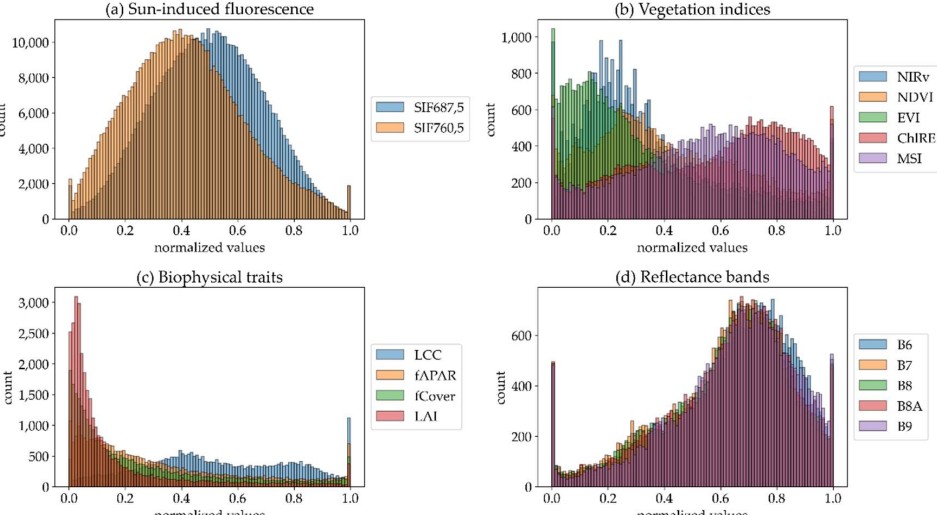

**Figure 6.** (**a**) Histogram of normalized values for sun-induced fluorescence; (**b**) Vegetation indices; (**c**) Biophysical traits; (**d**) Reflectance bands. Notice that all the distributions (**b**–**d**) are skewed, compared to (**a**).

*3.2. Models' Performances*

3.2.1. Evaluation

In general terms, the Vis, biophysical traits, and S2 bands related to vegetation structure (i.e., NIRv, fAPAR, fCover, LAI, B7 (red edge)) predicted the best SIF spatial heterogeneity (Figure 7). Regarding the methods, ensemble decision trees (Figure 7a) and spatial heterogeneity coefficient using SCL-5 (Figure 7e) were the best-performing methods ($SIF_{760,300}$ $R^2$ > 0.8 and $SIF_{687,300}$ $R^2$ > 0.6). Interestingly, when the spatial heterogeneity coefficient method was implemented with 15 classes (Figure 7d) instead of 5 (Figure 7e), the $R^2$ decreased to ~0.5 for $SIF_{760,300}$ and between 0.1–0.2 for $SIF_{687,300}$. The standard deviation method (Figure 7b), despite its simple implementation, casted $R^2$ > 0.6 for $SIF_{760,300}$ and $R^2$ > 0.1–0.4 for $SIF_{687,300}$ when NIRv, fCover, LAI and B7 were used as predictors, showing similar results to the spatial heterogeneity coefficient SCL-15 approach. Finally, the Local Moran's I (Figure 7c) and normalized entropy (Figure 7f) methods achieved the worst results, with $SIF_{760,300}$ $R^2$ < 0.3 and $SIF_{687,300}$ $R^2$ < 0.1. It is worth noting that the spatial heterogeneity was better predicted for $SIF_{760,300}$ than for $SIF_{687,300}$.

Despite its performance, the suitability of the spatial heterogeneity coefficient SCL-5 must be reconsidered. In this study area, the method relied on a categorization of five classes, i.e., water, cropland, pasture, forest and other. Since it is an agricultural area, most of the pixels are classified as cropland (Figure 4). Based on the formulation of the spatial heterogeneity coefficient (Table 3), this translates into zero heterogeneity, as the majority of S2-based sub-pixels within a 300 × 300 m FLEX pixel belong to the same land cover class. However, the same land cover class does not imply the same optical properties or fluorescence emission, as assumed by the method. Therefore, considering the potential deviation of these assumptions from the processes taking place in the scene, we decided to exclude the spatial heterogeneity coefficient SCL-5 from the list of suitable methods. For the subsequent analyses, we focused on the ensemble decision trees, spatial heterogeneity coefficient SCL-15, and standard deviation methods. For these models, we chose the best-performing predictors: NIRv, fAPAR and B7 (red edge). Note that we decided to use fAPAR instead of fCover because they are highly correlated [60], and in this study similar results were found in both biophysical traits when describing $SIF_{760}$ spatial heterogeneity. Furthermore, fAPAR presented slightly better results than fCover for $SIF_{687}$ for ensemble decision trees and for spatial heterogeneity coefficient SCL-15 methods.

3.2.2. Outliers' Spatial Distribution

When comparing the spatial heterogeneity metric values computed for the reference (SIF) and model predictor pixels using the RMSE, we found that the pixels with the highest RMSEs turned out to be the same for many methods. Figure 8 shows the number of times each pixel was classified as an outlier for all selected models for $SIF_{760,300}$ (Figure 8a), $SIF_{687,300}$ (Figure 8b), and the sum of both (Figure 8c). Outliers were more often located in the scene borders. However, this was not related to the absence of the missing ('NaN') high-resolution values, which was prevented by cropping the pixels on the edge of the image in the preprocessing steps.

Pixels with the ID 124 and 166 were outliers for most methods in both SIF bands; therefore, we decided to further investigate the spatial distribution of $SIF_{760,5}$, $SIF_{687,5}$, $NIRv_{20}$, $fAPAR_{20}$, $B7_{20}$ and the scene classification maps with 5 and 15 classes (Figure 9). For pixel ID 166, its spatial distribution is shown for SIF (Figure 9a,b), as well as for NIRv (Figure 9c), fAPAR (Figure 9d) and B7 (Figure 9e).

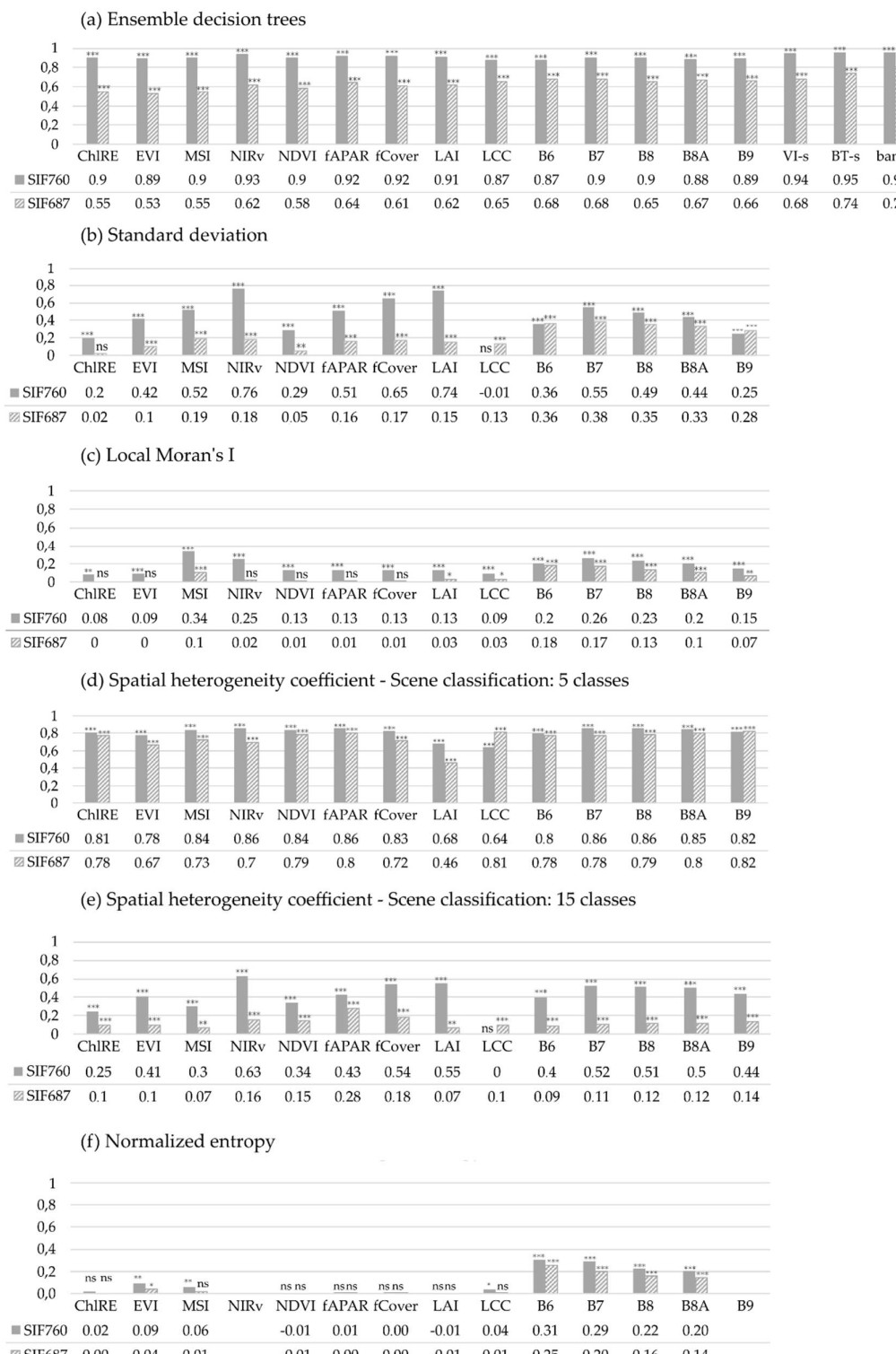

**Figure 7.** Square of Pearson's correlation coefficient between reference $SIF_{760}$ and $SIF_{687}$ heterogeneity and predictors' heterogeneity: Sentinel-2 derived vegetation indices, biophysical traits, reflectance bands and their stacks using the following methods: (**a**) Ensemble decision trees; (**b**) Standard deviation; (**c**) Local Moran's I; (**d**) Spatial heterogeneity coefficient using scene classification with 5 classes (SCL-5); (**e**) Spatial heterogeneity coefficient using scene classification with 15 classes (SCL-15); (**f**) Normalized entropy. *** $p$-value $\leq$ 0.001; ** $p$-value $\leq$ 0.01; * $p$-value $\leq$ 0.05, ns—$p$-value $> 0.05$.

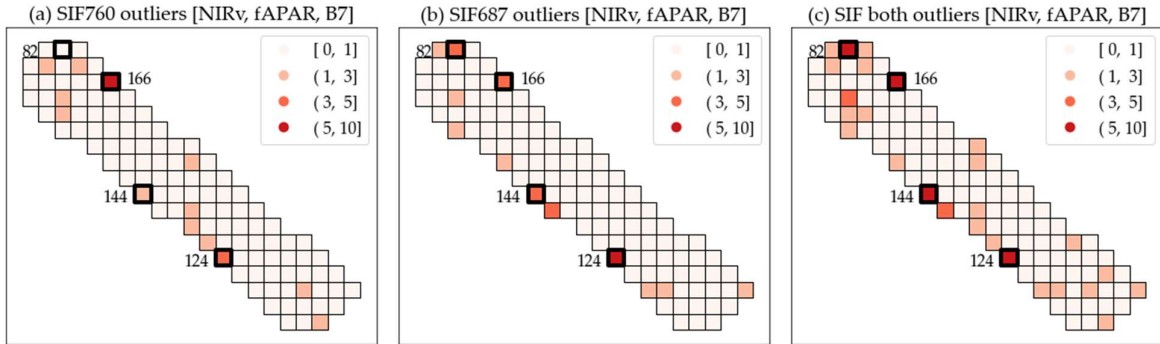

**Figure 8.** The number of times a pixel (pixel ID shown as numbers next to pixels) was considered an outlier (top 6 RMSE) by ensemble decision trees, spatial heterogeneity coefficient with SCL-15 and standard deviation methods using the most important predictors from each category (i.e., NIRv—vegetation index category, fAPAR—biophysical trait category, B7—reflectance band category) for (**a**) $SIF_{760,300}$, (**b**) $SIF_{687,300}$ and (**c**) the sum of counts for both $SIF_{760,300}$ and $SIF_{687,300}$. The maximum possible count for (**a**,**b**) is 9 (three models, three predictors), for (**c**) 18.

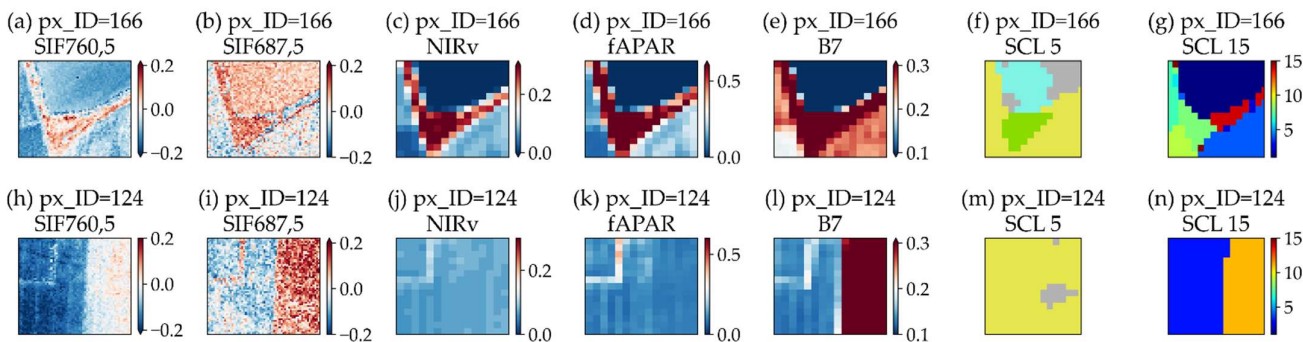

**Figure 9.** Top two outlier pixels (ID 166 and 124) with input data for (**a**,**h**) $SIF_{760,5}$; (**b**,**i**) $SIF_{687,5}$; (**c**,**j**) NIRv; (**d**,**k**) fAPAR; (**e**,**l**) B7; (**f**,**m**) SCL-5 classes; (**g**,**n**) SCL-15 classes. SIF units are W m$^{-2}$ sr$^{-1}$ μm$^{-1}$, NIRv and fAPAR are unitless; and B7 is in sr$^{-1}$. Classes for SCL-5 are water (cyan), cropland (olive), pasture (green), and other (gray), as in Figure 4. Classes for SCL-15 are discrete values and represent spectral rather than land cover classes.

Finally, regarding the scene classification maps, for pixel 166, the supervised classification resulted in four different classes—crops, pasture, water and other (Figure 9f)—while pixel ID 124 resulted in two different classes (mostly crops and a smaller area of other (unidentified) classes) (Figure 9m). Unsupervised SCL-15 reflected the spatial structure of SIF in pixels better than SCL-5, resulting in clearer boundaries of objects in the image. A clear pattern is observed, with higher values in the center of the pixel (V-shape) and lower values in the surrounding areas. Regarding the scene classification maps, for pixel 166, supervised classification resulted in four different classes—crops, pasture, water and other (Figure 9f)—while pixel ID 124 resulted in two different classes (mostly crops and a smaller area of other (unidentified) classes) (Figure 9m). Unsupervised SCL-15 reflected the spatial distribution of SIF pixels better than SCL-5, resulting in clearer boundaries of objects in the image. For pixel ID 124, a common pattern is observed for $SIF_{760,5}$ and $SIF_{6875,5}$ values (Figure 9h,i), where approximately half of the pixels have SIF values greater than 0.2 W m$^{-2}$ sr$^{-1}$ um$^{-1}$ and the other half have SIF values close to zero or negative. Notably, $SIF_{687,5}$ shows higher values than $SIF_{760,5}$ for both pixels. Regarding the spatial distribution of NIRv (Figure 9j) and fAPAR (Figure 9k) for pixel ID 124, a homogeneous distribution across pixels was observed. Interestingly, the spatial distribution of B7 (Figure 9l) follows the pattern of SIF.

### 3.2.3. Best-Performing Models

The methods that best described SIF heterogeneity included ensemble decision trees, standard deviation and the spatial heterogeneity coefficient using SCL-15. The spatial heterogeneity coefficient using the five-classes method was excluded because many pixels were classified exclusively as cropland, making the heterogeneity values zero (see section Models' Performances for a more detailed explanation). For all the methods, the NIRv index provided the best characterization of SIF spatial heterogeneity; therefore, the performance of each method is shown based on this index (Figure 10). Models with $SIF_{687,300}$ did not perform as well and are not included in this section but are included in Appendix C (Figures A1–A3).

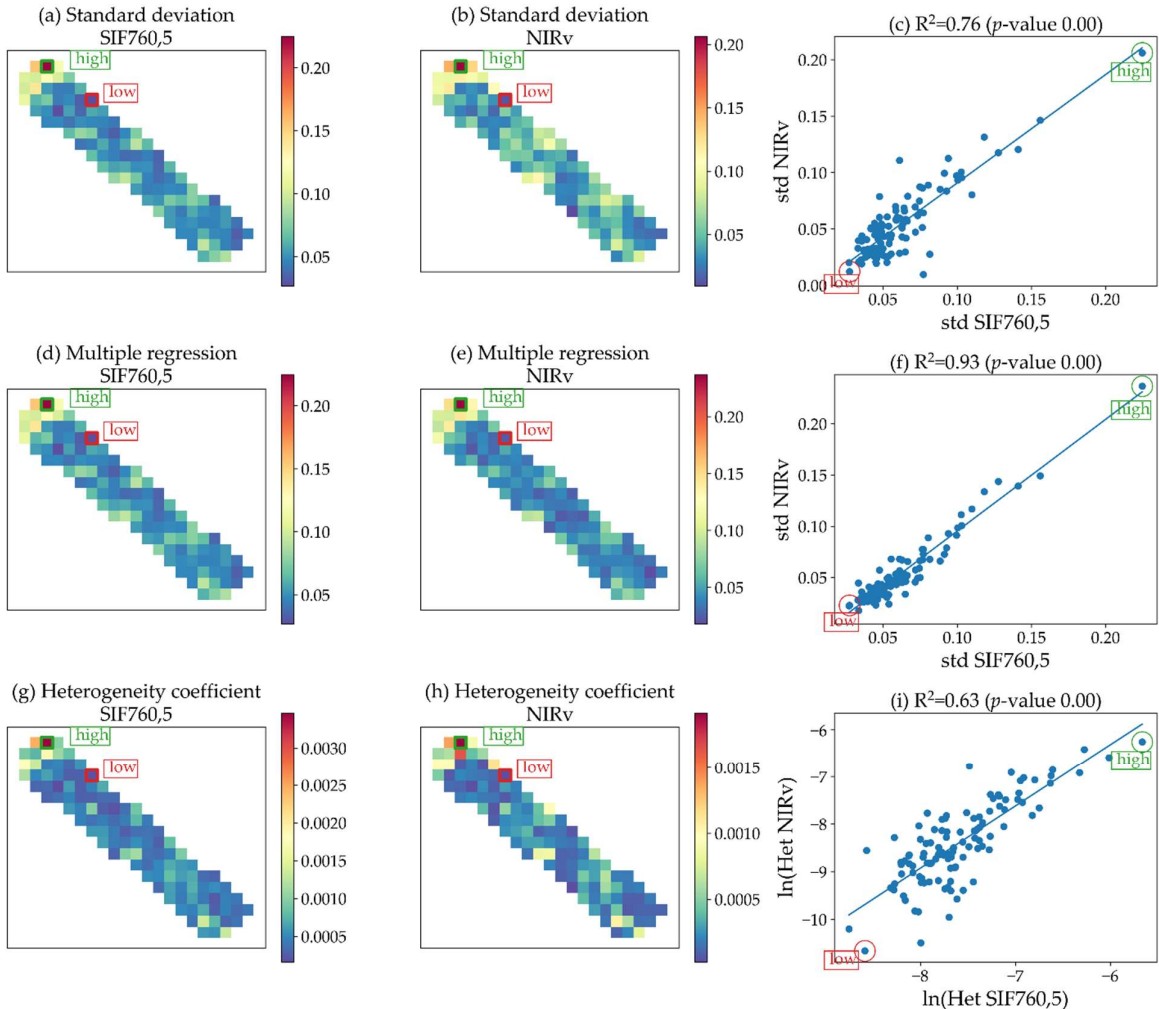

**Figure 10.** Heterogeneity maps (300 × 300 m) for standard deviation, ensemble decision trees and spatial heterogeneity coefficient SCL-15 methods; (**a,d,g**) reference $SIF_{760}$; (**b,e,h**) best predictor NIRv; (**c,f,i**) scatter plots with lowest (green circle) and highest (red circle) heterogeneity pixels highlighted.

Figure 10 shows the distribution of spatial heterogeneity coefficient values of the study area based on $SIF_{760,5}$ reference data (Figure 10a,d,g) and predictor NIRv (Figure 10b,e,h) for all three selected methods. Lower values indicate low heterogeneity and higher values indicate high heterogeneity. There is a significant linear relationship between the reference and predictor SIF spatial characterization for standard deviation ($R^2 = 0.76$, $p < 0.001$, Figure 10c), ensemble decision trees ($R^2 = 0.93$, $p < 0.001$, Figure 10f) and spatial heterogeneity coefficient ($R^2 = 0.63$, $p < 0.001$, Figure 10i).

From the scatterplots in Figure 10, pixels with high (highlighted in green) and low (highlighted in red) spatial heterogeneity have been magnified in Figure 11.

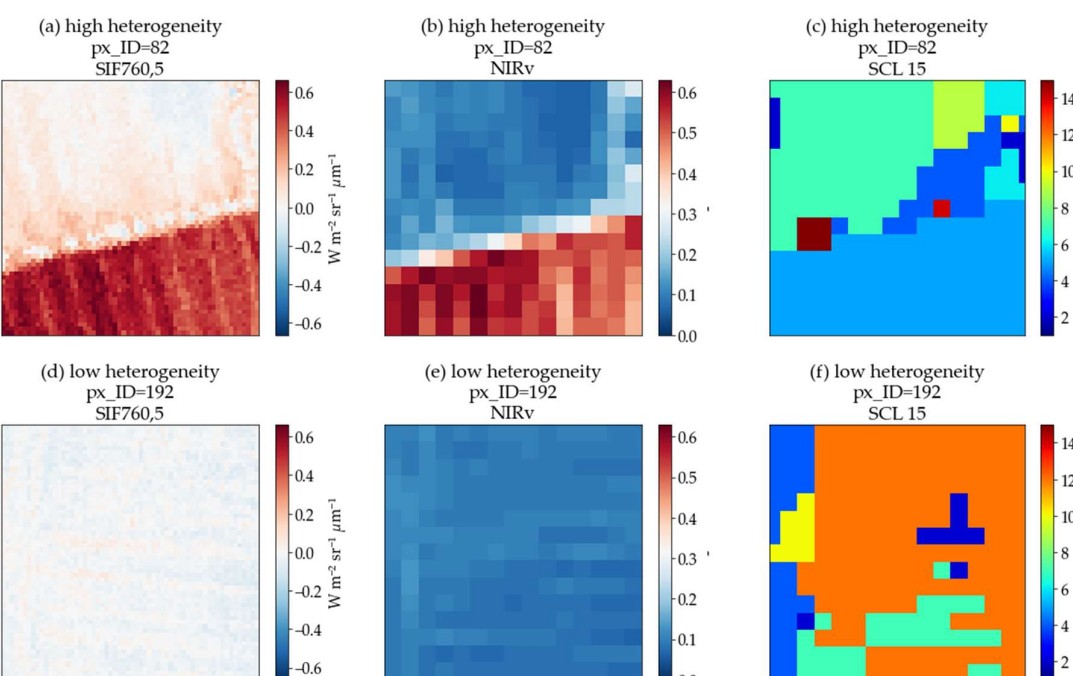

**Figure 11.** Highest and lowest heterogeneity pixels from best-performing models (standard deviation, spatial heterogeneity coefficient using 15 classes, ensemble decision trees) with input data shown for (**a**,**d**) $SIF_{760,5}$; (**b**,**e**) $NIRv_{20}$; (**c**,**f**) scene classification map 15 classes.

For all methods, the pixel with the highest heterogeneity (i.e., ID 82) was located in the northwestern upper part of the image and was consistently characterized by heterogeneous $SIF_{760,5}$ and $NIRv_{,20}$ values with eight land cover classes (Figure 11a–c). At pixel ID 82, higher values were observed grouped in the lower right part of the pixel for $SIF_{760,5}$ and $NIRv_{,20}$ maps. The scene classification map had nine different classes, with the lower part of the image fully following the SIF and NIR patterns, while the upper parts were less visually aligned. A less heterogeneous pixel, ID 192, was also observed in the northern edge of the image with homogeneous $SIF_{760,5}$ and $NIRv_{,20}$ maps (Figure 11d,e), which had low values. The scene classification map had five different classes, one of which dominated the image (orange color). In the left part of the pixel, a contrasting edge can be seen on the SCL map, which is not as pronounced on the $SIF_{760,5}$ and $NIRv_{20}$ maps.

## 4. Discussion

In this study, ensemble decision trees, standard deviation and the spatial heterogeneity coefficient using scene classification with 15 classes provided the best estimates of SIF intrapixel heterogeneity (Figure 10, p. 17). Between the S2 bands, vegetation indices, and biophysical traits, previously filtered with the SSIM (Figure 3, p. 7), SIF heterogeneity was best characterized when the NIRv index, fCover and LAI were used as input data (Figure 7, p. 14). The use of the NIRv index, defined as the product of NIR reflectance and normalized vegetation index (NDVI), isolates the vegetation signal, enhances structural properties by multiplying NDVI by NIR reflectance, and eliminates the mixed pixel problem that occurs when using NDVI [43]. Several studies have shown a significant positive correlation between NIRv and fAPAR when background effects are significant [61]. Since SIF is first-order driven by changes in APAR [23], and since we are analyzing a single time image in this study, it would be expected to be linearly correlated with SIF, thus explaining the heterogeneity of SIF. However, Dechant et al. [62] have shown that NIRv tends to saturate at high SIF values when different seasonal conditions and/or ecosystem vegetation fractional cover are integrated to linearly correlate NIRv and SIF. Therefore, further studies should be performed to investigate the correlation between SIF and NIRv when different stress

levels and/or illumination conditions (i.e., sunlit shaded areas due to different vegetation fraction cover) are considered.

The ensemble decision trees method provided the best results, but also, unlike other methods, it combined $SIF_{760,300}$ and $SIF_{687,300}$ with the different S2 bands, VIs and BTs considered in this study, which significantly improved the model performance. However, caution should be exercised when extrapolating these results to other studies, such as vegetation dynamics monitoring. When downscaling SIF from $300 \times 300$ to $20 \times 20$ m, we assume the same fluorescence response at each $20 \times 20$ sub-pixel englobed in a FLEX pixel, which is rather unexpected due to the dynamic response of fluorescence and consequently could lead to a misinterpretation of the vegetation status.

For the spatial heterogeneity coefficient method, heterogeneity is determined by both the number of land cover types (Shannon's entropy) and the range of values within each type (class variance). When applied to a homogeneous pixel with one land cover type, the heterogeneity is automatically zero, so this method is limited by the number of classes in the scene classification. When using the supervised scene classification map, we hypothesize that five classes were not enough to accurately classify all sub-pixels (i.e., it included many sub-pixels in the crop class when they should have been classified as pasture or created as a separate class). Many pixels were classified in only one class (i.e., crop), so both the heterogeneity of the SIF and the predictors were zero, resulting in a high correlation model. Alternatively, unsupervised scene classification can provide many classes based on statistical differences between the spectral properties of the surfaces, which are independent of subjective interpretations. However, the number of classes that can best characterize SIF intrapixel heterogeneity, bearing in mind that more classes inherently lead to higher heterogeneity, remains unclear and might be scene dependent. For example, Zhao and Fan [31] used only three land cover types in their study, which aimed to quantitatively express LAI spatial heterogeneity. They observed that ratios and increases in the number of land cover types lead to changes in the heterogeneity coefficient and suggested it should be investigated further. These arguments lead to the recommendation to use k-means clustering for SCL, which better distinguishes natural gradients and class boundaries, and later to further investigate what is the optimal number of clusters to characterize a given area.

The standard deviation method is based on the arithmetic average (mean) and is intended to identify unobserved spatial heterogeneity in pixels [63]. The standard deviation was previously used as a measure of the spatial variability of soil moisture in Li and Rodell [64], while Riera et al. [65] used the standard deviation of NDVI as an expression of vegetation heterogeneity. Spatial heterogeneity expressed as the standard deviation of $SIF_{760,5}$ showed a significant linear correlation with the NIRv index ($R^2 = 0.76$, *p*-value < 0.001) (Figure 10c), suggesting that it was successful in capturing the spatial heterogeneity of the fluorescence emitted at 760 nm in this study. Regarding $SIF_{687}$, only the standard deviations of S2-B6, S2-B7 (both red-edge) and S2-B8 (NIR) were able to explain the ~40% of $SIF_{687}$ spatial heterogeneity . A similar approach was implemented in Rossini et al. [1], in which they used the absolute deviation from the mean between ground $SIF_{760}$ observations and medium-resolution $SIF_{760}$ pixels (300 m) to determine the optimal sampling strategy (i.e., number of sampling points) to characterize a FLEX-based pixel over an agricultural area. They concluded that between 3 and 13.5 sampling points are required to characterize the average SIF value of a monoculture field at a FLEX-based pixel resolution.

Normalized entropy and Local Moran's I models contributed less to the characterization of SIF spatial heterogeneity (Figure 7). The key issue with the normalized entropy method lies in its approach. Unlike the spatial heterogeneity coefficient, which clarifies how different land cover classes contribute to pixel values, the normalized entropy method utilizes the actual data values of each pixel as a relative contribution to the broader distribution of sub-pixel values. Hence, it does not detect smaller or larger differences in the surrounding pixels [66]. We hypothesize that normalized entropy, based on Shannon's approach, did not work in our study because it was mostly dominated by crops with similar SIF, VIs, and BT values. The concept of entropy introduced by Shannon [59] was compared

to Rao's Q entropy [67] in [66,68] to characterize the spatial heterogeneity of an image. In these studies, Tagliabue et al. [68] found that using Rao's Q entropy to obtain SIF and NDVI heterogeneity was effective as a measure of functional diversity in forests. However, they conclude that the ESA-FLEX pixel is too coarse to assess functional diversity. They suggest using the increasingly available hyperspectral imagery to downscale the SIF signal using machine learning or unmixing approaches. Furthermore, Doxa and Prastacos [66] compare Shannon's entropy with Rao's quadratic entropy, stating that the former tends to overestimate environmental heterogeneity, while the latter highlights pixels with significantly different values. Therefore, Rao's entropy could be tested as an alternative to normalized entropy. Local Moran's I has a potential in making distinct clusters based on pixel values and their spatial arrangement, where statistically insignificant and single-member clusters (diamonds and doughnuts) potentially express heterogeneity. Like normalized entropy, this approach in our study did not produce good results.

Regarding the strengths and weaknesses of the proposed methods, ensemble decision trees generally produce highly accurate predictions because they reduce overfitting by averaging multiple decision trees, making them more reliable for a wide range of datasets. Random forests can also handle missing values without requiring imputation, making them versatile for real-world data with incomplete information. They can capture complex, nonlinear relationships between characteristics and the target variable that linear models may struggle with. On the other hand, random forests are generally unsuitable for extrapolation because they tend to make predictions based on patterns observed in the training data. Predictions outside the range of the training data may be less reliable. The advantage of the standard deviation is that it is simple and easy to calculate. However, the standard deviation method can smooth the effect of a smaller heterogeneous area surrounded by a larger homogeneous area within a sub-pixel. Furthermore, we expect the variability of the vegetation index to be proportional to the variability of the SIF, whereas in reality the SIF may be more heterogeneous (for physiological reasons) if the vegetation index remains homogeneous. The spatial heterogeneity coefficient combines interclass and intraclass heterogeneity and can capture variations of values within the same land class and weigh them according to their area within a sub-pixel. It uses scene classification maps that can be easily generated with S2 data using k-means clustering, although it remains to be explored how to choose the number of clusters for different study areas.

Observing the maps of the study areas, two clear patterns could be observed in the northern (pasture as a part of the dairy farm) and central (irrigated corn crop) areas with higher values of SIF, VI, BT (pattern 1) compared to the rest of the image (pattern 2) (Figure 5). According to the FlexSense final report (ESA contract no. 4000125402/18/NL/NA), the corn field reached full canopy cover at the time of the HyPlant overpass, which is consistent with the observed VI and BT maps, e.g., MSI shows lower values indicating higher leaf water content, which is expected in irrigated areas. In addition, patterns may also be explained by differences in management systems, such as available resources—i.e., nutrients from fertilizers—pastures would have some amount of manure depending on the type of livestock, and crops could be managed with artificial fertilizers. The report also states that from the month of June onward, the circular crop was surrounded by dry grasslands or soils. Interestingly, the patterns described above are mainly consistent between $SIF_{760,5}$ and the VI, BT and S2 reflectance band images in the study, but a higher variability is observed when looking at the $SIF_{687,5}$ map (Figure 5a). We hypothesize that this could be due to the higher retrieval uncertainty at $SIF_{687,5}$ when SIF is retrieved in pixels with high intrapixel heterogeneity, i.e., low vegetation fractional cover [24], as well as for the lower signal-to-noise ratio of the HyPlant instrument [69,70], which alters the measured spectral reflected radiance shape and increases the fluorescence retrieval uncertainty. Interestingly, the pixels identified as outliers support the hypothesis just described, for example, pixels ID166 and ID124 present low NDVI and LAI values (Figure 5), suggesting low fractional cover and consequently making the retrieval more prone to cast bias.

In summary, models with $SIF_{760}$ consistently performed better than those with $SIF_{687}$. While canopy $SIF_{687}$ is dominated by chlorophyll re-absorption within the leaves and canopy, the $SIF_{760}$ signal is primarily affected by scattering owing to leaf and canopy structural properties as well as solar-and-observation angle [71–73]. In this study, S2-based images were used to estimate different VI and BT, but due to the S2 configuration, only structure-related VI and BT could be retrieved (i.e., NIRv, fAPAR, fCover, LAI), which explains why the proposed model better characterizes the spatial heterogeneity of $SIF_{760}$, but fails to characterize the weaker—and most dynamic—fluorescence emission at 687 nm [22,74]. Furthermore, the characterization of SIF intrapixel heterogeneity analyzes a single scene at a single point in time, so it is expected that the static vegetation structure traits will drive the field variability. In a time series analysis, however, the changes in environmental conditions (i.e., PAR, water and nutrient availability) would instead be captured by the dynamic vegetation traits, such as APAR and/or NPQ [75].

## 5. Conclusions

SIF is a highly dynamic signal, and changes in environmental growing conditions (i.e., resource availability, light conditions, vegetation fractional cover or land cover) can bias the interpretation of satellite SIF estimates. In the context of the SENSECO COST Action, the HyPlant dataset acquired over an agricultural area during the ESA FlexSense campaign (ESA contract no. 4000125402/18/NL/NA) was used to characterize the spatial heterogeneity of a FLEX pixel based on S2 reflectance-based predictors and using different patch mosaic and direct heterogeneity methods. The application of different methods to describe the spatial heterogeneity of sun-induced fluorescence provided interesting insights, particularly for the use of standard deviation, the spatial heterogeneity coefficient and ensemble decision trees using machine learning algorithms. Among all methods, the structured related NIRv index explained SIF heterogeneity the best, compared to other vegetation indices (i.e., NDVI, ChlRE, EVI), biophysical traits (i.e., fAPAR, fCover, LCC, LAI) or S2 reflectance bands. The methods proposed in this study are simple to implement and could be used to develop cal/val sampling protocols, define quality labels based on the spatial variability of a FLEX pixel (i.e., high/low heterogeneity), and improve SIF retrieval in pixels labeled as highly heterogeneous. We are aware that the results of this study are limited to homogeneous agricultural sites. Therefore, based on the results of the current study, future work should include the application of the proposed methods to a larger dataset, such as other regions with different ecosystems (i.e., forest, grassland, boreal and/or tropical forest) to then determine which combination of method and VIs or biophysical traits should be implemented in each ecosystem type. If our results are confirmed, our study would be a precursor for a SIF pixel heterogeneity product that could be implemented in the FLEX ground segment processing chain.

**Author Contributions:** Conceptualization, M.P.C.-M., M.C. and N.J.; methodology, M.P.C.-M., M.C., N.J., E.P., E.T., I.H.-S., F.P. and S.B.; formal analysis, E.P.; data curation, N.J., E.P. and M.C.; writing—original draft preparation, N.J. and M.P.C.-M.; writing—review and editing, N.J., M.P.C.-M., J.P.-L., E.T., I.H.-S., S.V.W., G.K., B.S., T.L. and H.K.; visualization, E.P. and N.J.; supervision, M.P.C.-M.; visualization, E.P. and N.J.; supervision, M.P.C.-M. All authors have read and agreed to the published version of the manuscript.

**Funding:** The paper was supported by a Virtual Mobility grant as a part of SENSECO, project launched in the framework of the COST Action CA17134.

**Institutional Review Board Statement:** Not applicable.

**Informed Consent Statement:** Not applicable.

**Data Availability Statement:** The data presented in this study are available on request from the corresponding author.

**Acknowledgments:** We gratefully acknowledge the financial support of the European Space Agency (ESA) for airborne data acquisition within the framework of the FLEXSense campaign (ESA/Contract No. 4000125402/18/NL/NA). Nela Jantol is supported by a career development project for young re-

searchers of the Croatian Science Foundation (Contract No. DOK-2020-01-9841). Shari Van Wittenberghe and M.Pilar Cendrero-Mateo are currently funded by the European Research Council (ERC) under the ERC-2021-STG PHOTOFLUX project (grant agreement 101041768). M.Pilar Cendrero-Mateo received funding from the Ministry of Science and Innovation of the Spanish Government (Grant no. IJC/2018/038039/I). Marco Celesti is funded by the Living Planet Fellowship of the European Space Agency (ESA/Contract No. 4000125442/18/I-NS). Javier Pacheco-Labrador is funded by the ESA Living Planet Fellowship IRS4BEF 'Integrated Remote Sens. for Biodiversity-Ecosystem Function' (C.N. 4000140028/22/I-DT-lr).

**Conflicts of Interest:** The authors declare no conflict of interest.

## Appendix A

**Table A1.** Detailed explanations for each method used for expressing heterogeneity and the range of heterogeneity values.

| Method Name | Heterogeneity Values |
| --- | --- |
| Local Moran's I | Range from 0 to 1, where 0 corresponds to low heterogeneity and 1 to high heterogeneity |

We quantified the fluorescence heterogeneity at FLEX resolution as the fraction of pixels belonging to a single-pixel cluster or not assigned to any cluster. For each $300 \times 300$ m pixel, we independently clustered S2-R$_{20}$ and SIF$_{\lambda,5}$ using the Local Moran's I method [57] implemented in the function Moran_Local() from the ESDA (Exploratory Spatial Data Analysis) package from the PySAL Python library [76]. The approach classifies pixels into four classes: diamond (a single high value among low values), doughnuts (a single low value among high values), hotspot (a high value among high values), and cold spot (a low value among low values); the first two are single-pixels classes. The classification is conducted based on the spatial autocorrelation Moran's I metric whose statistical significance is defined using permutations (bootstrap). In this way, the pixels that were not significantly assigned to any of the former categories constituted a new "heterogeneous" class. FLEX pixel heterogeneity was then computed as the fraction of diamond, doughnut, and non-significant pixels within each $300 \times 300$ m FLEX pixel.

| Method Name | Heterogeneity Values |
| --- | --- |
| Spatial heterogeneity coefficient | High values determine higher heterogeneity and lower values lower heterogeneity. The lowest possible value is 0, when all pixels belong to one class (homogeneous). The highest value is limited by the range of the values and the number of land cover types, 0.0053 $W^2\ m^{-4}\ um^{-2}\ sr^{-2}$ for SIF760, 0.55 $m^4\ m^{-4}$ for LAI |

Heterogeneity was quantified as a function of land-use cover variability within each FLEX 300 m pixel, using the spatial heterogeneity coefficient described in [31].
Formula for spatial heterogeneity coefficient ($C_{sh}$):

$$C_{sh} = \sum_{i=1}^{N} p_i \times E_i \times \sigma_i^2$$

$$= -\sum_{i=1}^{N} P_i^2 \ln(p_i) \frac{\sum_{m=1}^{n_i} (x_m - \mu)^2}{n_i}$$

N—total number of land cover classes of each sub-pixel included in a pixel;
$x_m$—m-th pixel value, which is included in the i-th land cover class;
$\mu$—mean value of the total sub-pixels that are included in one pixel;
$n_i$—total number of sub-pixels included in the i-th type;
$p_i$—fraction of the i-th land cover class in a pixel.
One FLEX pixel could contain more than one land cover class; such a class is represented by sub-pixels (5 m for SIF and 20 m for S2 metrics). Heterogeneity combines class variance with information entropy. Class variance ($\sigma_i^2$) is the difference in sub-pixels reflecting intraclass (difference in growth conditions for the same vegetation type, i.e., different canopy densities) and interclass (i.e., land cover class patchiness) heterogeneity. Information entropy ($E_i$) or class frequency explains how much each land cover class ($p_i$) contributes to the pixels and is expressed as a fraction of a specific land cover class in a pixel multiplied by its natural logarithm. Two scene classification maps with 5 (SCL-5) and 15 classes (SCL-15) were produced using supervised and unsupervised approaches, respectively. Information from the Urban Atlas layer was used for creating a simpler SCL-5 (containing five classes defined as crops, pasture, water, forest, other) using semi-automatic classification plugin [60] on S2 bands in the QGIS environment ("QGIS Geographic Information System," 2021) (Figure 4A). Another SCL-15 map was produced using k-means clustering on the S2 dataset in SAGA with 15 clusters (the same number of land cover types for Braccagni image as in the Urban Atlas layer). Both maps were smoothed out with a $3 \times 3$ mode (majority) kernel.
The spatial heterogeneity coefficient was calculated for every 300 m pixel using land cover frequencies and land cover class variances from scene classification maps.

| Method Name | Heterogeneity Values |
| --- | --- |
| Standard deviation | High values determine higher heterogeneity and lower values lower heterogeneity. The range depends on the range of the predictor, i.e., from 0.03 to 0.22 $W\ m^{-2}\ um^{-1}\ sr^{-1}$ for SIF$_{760}$, 0.05 to 2.52 $m^2\ m^{-2}$ for LAI. |

The standard deviation is a measure of how dispersed the data are in relation to their mean. Riera et al. [61] used the standard deviation of the *NDVI* as an expression of vegetation heterogeneity; moreover, Li and Rodell [62] used it as a measure of spatial variability of soil moisture.

| Method Name | Heterogeneity Values |
| --- | --- |
| Ensemble decision trees | High values determine higher heterogeneity and lower values lower heterogeneity. This method converted predictor values to SIF values; therefore, the range was always from 0.03 to 0.22 $W\ m^{-2}\ um^{-1}\ sr^{-1}$ for SIF$_{760}$ and from 0.05 to 0.11 $W\ m^{-2}\ um^{-1}\ sr^{-1}$ for SIF$_{687}$ |

We assessed the capability of four different machine learning algorithms to predict SIF$_{\lambda,20}$ as a function of SIF$_{\lambda,300}$, and R$_{20}$: eXtreme Gradient Boosting, Random Forests, support vector machines, and neural networks. The imagery was randomly split into training and validation subsets based on the finest resolution. For training models, 20% of the data (6800 samples) were used for computation economy. We used a k-fold ($k = 5$) cross-validation approach to assess each algorithm's performance. Random Forests [58] was the most accurate algorithm. Thus, we made use of this approach to upscale SIF from the FLEX to the S2 spatial resolution.

| Method Name | Heterogeneity Values |
| --- | --- |
| Normalized Entropy | Normalized entropy ranging from 0 to 1, where 1 corresponds to low heterogeneity and 0 to higher heterogeneity. Probability values pi are expressed for sub-pixel locations i inside the FLEX pixel, and their values correspond to the sub-pixel values in S2 or F |

$$E = \sum_{i=0}^{N} (p_i \times \log_2(p_i))$$
$$p_i = \frac{x_i}{\sum_{i=0}^{N} x_i}, \text{ where } x_i - \text{pixel value}$$
$$E_{max} = E_{uniform}, \text{ where } p_i = \frac{1}{N}$$
$$\frac{E}{E_{max}} - \text{heterogeneity}$$

Heterogeneity was quantified using the concept of entropy [64] that measures the average information content. Entropy is maximized when every sub-pixel within the $300 \times 300$ m FLEX pixel contains the same value (uniform probability distribution, no heterogeneity). Thus, within a 300 m pixel, the maximum possible entropy value for 225 20 m sub-pixels (S2 products) is 7.81 (all $p_i = 1/225$) and for 3600 5 m sub-pixels it is 11.81 (all $p_i = 1/3600$). For each pixel at FLEX resolution (SIF$_{\lambda,300}$), we calculated the entropy using each dataset SIF$_{\lambda,5}$, S2-VI$_{20}$, S2-BT$_{20}$, and S2-R$_{20}$. We normalized the entropy by the entropy of the uniform distribution ($E_{max}$ with N = sub-pixels in a FLEX pixel). This custom "normalized entropy" function was passed as an additional parameter to the python module rasterstats [77].

## Appendix B

Two additional methods were attempted to describe the spatial heterogeneity of a FLEX pixel but were unsuccessful in retrieving heterogeneity. These methods are described below.

**Table A2.** Additional methods used but failed to characterize SIF heterogeneity.

| Method Name | Heterogeneity Definition | Predictors | Heterogeneity Values |
|---|---|---|---|
| Cluster entropy | Uncovered sub-pixel information by aggregating patterns with a similar distribution. | $S2\text{-}VI_{20}$ $S2\text{-}BT_{20}$ $S2\text{-}R_{20}$ | Uncovered sub-pixel information by aggregating patterns with a similar distribution. |

A set of spatial patterns were extracted from $S2\text{-}VI_{20}$, $S2\text{-}BT_{20}$ and $S2\text{-}R_{single,20}$ by means of a clustering approach, aggregating patterns with a similar distribution. Analogously, patterns from SIF at 5 m pixel resolution are also extracted. This allows us to compare the uncovered sub-pixel information for both co-registered datasets.

For this method, $300 \times 300$ m patches from SIF and S2 data were extracted, and considering that the SIF spatial resolution is 5 m, each patch contains $60 \times 60$ sub-pixels or 3600 pixels, whereas for FLEX patches are represented by $15 \times 15$ or 225 pixels. All the patches containing missing values were discarded, leaving a total of 110 patches for SIF and 114 patches for S2. Nevertheless, only 104 patches of S2 and SIF overlapped and were used for the comparison. We used the $SIF_{760}$ and $SIF_{687}$ as reference data and the S2 reflectance bands and its derived indices as predictors. A Gaussian mixture model (GMM) clustering algorithm [78] grouped the 300 m patches of $F_{\lambda,5}$ into k = 3 clusters. $S2\text{-}VI_{20}$, $S2\text{-}BT_{20}$ and $S2\text{-}R_{single,20}$ were clustered into k = 4 groups as they showed higher heterogeneity than the patches of $F_{\lambda,5}$. Since cluster labeling was arbitrary, clusters were relabeled from 1 to 3, allowing for comparing Sentinel-2 and $F_{\lambda,5}$ groups. The capability of the different S2 predictors to capture $F_{\lambda,5}$ on FLEX scales (300 m) was evaluated using confusion matrices.

| | Model fluorescence ($SIF_{760,20}$ and $SIF_{687,20}$) sub-pixels variance in a 300 $\times$ 300 FLEX pixel | $S2\text{-}VI_{20}$ $S2\text{-}BT_{20}$ $S2\text{-}R_{20}$ | |
|---|---|---|---|
| Fuzzy approach | | | |

Fuzzy modelling allows for building flexible weighted maps from several variables [79], which can be used as a predictor of a third variable once combined. It is a two-step method where the original variables ($VI_{20}$ or $BT_{20}$) are first transformed to the range [0, 1] using different "membership" functions. These are selected according to their expected or known relationship with the predicted variable ($SIF_\lambda$ in this case). Then, the transformed variables (membership values) can be combined through various operators [80]. The combined values can then be used as predictors of the variable of interest.

In this work, we applied fuzzy modelling to $VI_{20}$ and $BT_{20}$ separately to eventually predict $SIF_\lambda$. We selected the membership functions so that membership values positively correlated with SIF (Table A3).

Membership values computed from $VI_{20}$ or $BT_{20}$ variables were combined using the fuzzy overlay operator GAMMA since it has been reported to offer a balance between over and underestimation of fluorescence radiance [80,81].

$\mu\_\gamma \ (x) = ⟦[\mu\_SUM \ (x)]⟧ \ ^\gamma * \ ⟦[\mu\_PRODUCT \ (x)]⟧ \ ^{(1 - \gamma)}$

Then a linear model was fit using the integrated membership values as a predictor of $SIF_\lambda$:

$(F\_\lambda) ⊕ = b\_0 + b\_1 \ \mu\_\gamma \ (x)$

Fuzzy modelling was applied to 5 and 20 m spatial resolution data, and then these maps were gridded to 300 m pixels. Predicted and observed $SIF_\lambda$ and their intrapixel variability were assessed.

**Table A3.** Membership functions selected for each spectral index or vegetation parameter and the corresponding justifications according to the expected correlation with fluorescence radiance. Here, *x* stands for the input variable (predictor), *μ* for the transformed membership value, *m* for the mean, and *s* for the standard deviation. *a* and *b* are scaling factors of the mean and the standard deviation and were set to 1 in all the cases.

| HyPlant Derived VIs | Membership Functions | Equations | Justifications | Representing Traits | References |
|---|---|---|---|---|---|
| NDVI | Fuzzy MS Large | $\mu(x) = 1 - \frac{bs}{x - am + bs} \ if$ $x > am \ otherwise \ \mu(x) = 0$ | Positive correlation | Greenness Content | [82–84] |
| Chl-Red edge | | | | Red-edge position | [84] |
| EVI | Fuzzy Linear | $\mu(x) = \left\{ \frac{x - min}{max - min} \right\}$ | Weak correlation | Biomass | [84,85] |
| MSI | Fuzzy MS Small | $\mu(x) = \frac{bs}{x - am + bs} \ if$ $x > am \ otherwise \ \mu(x) = 1$ | Negative correlation | Canopy water stress | [84,86,87] |

## Appendix C

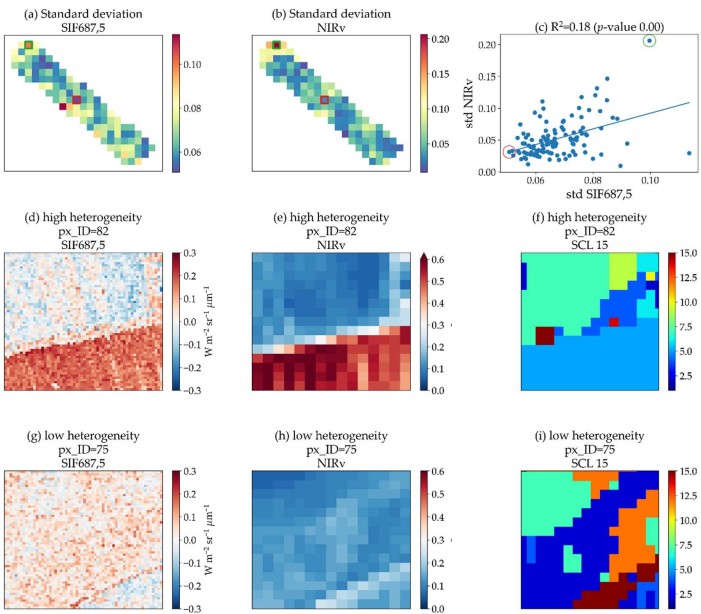

**Figure A1.** Heterogeneity maps (300 × 300 m) for the standard deviation method: (**a**) Reference SIF$_{687,5}$; (**b**) Best predictor NIRv; (**c**) Scatter plot with lowest (red circle) and highest (green circle) heterogeneity pixels highlighted; (**d**) 5 m pixel with high heterogeneity for SIF$_{687,5}$; (**e**) 20 m pixel with high heterogeneity for NIRv; (**f**) Scene classification with 15 classes for a pixel with high heterogeneity; (**g**) 5 m pixel with low heterogeneity for SIF$_{687,5}$; (**h**) 20 m pixel with low heterogeneity for NIRv; (**i**) Scene classification with 15 classes for a pixel with low heterogeneity.

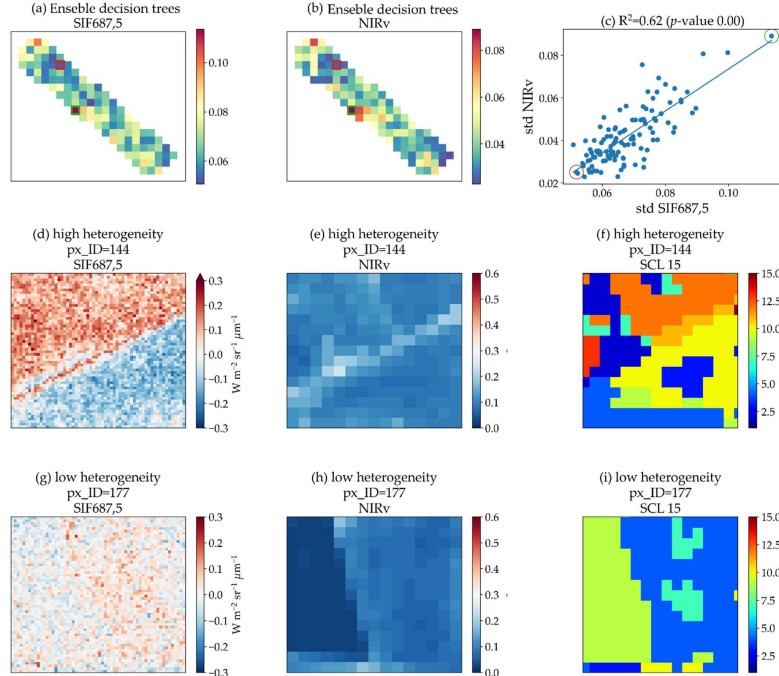

**Figure A2.** Heterogeneity maps (300 × 300 m) for the ensemble decision trees method: (**a**) Reference SIF687,5; (**b**) Best predictor NIRv; (**c**) Scatter plot with lowest (red circle) and highest (green circle) heterogeneity pixels highlighted; (**d**) 5 m pixel with high heterogeneity for SIF$_{687,5}$; (**e**) 20 m pixel with high heterogeneity for NIRv; (**f**) Scene classification with 15 classes for a pixel with high heterogeneity; (**g**) 5 m pixel with low heterogeneity for SIF$_{687,5}$; (**h**) 20 m pixel with low heterogeneity for NIRv; (**i**) Scene classification with 15 classes for a pixel with low heterogeneity.

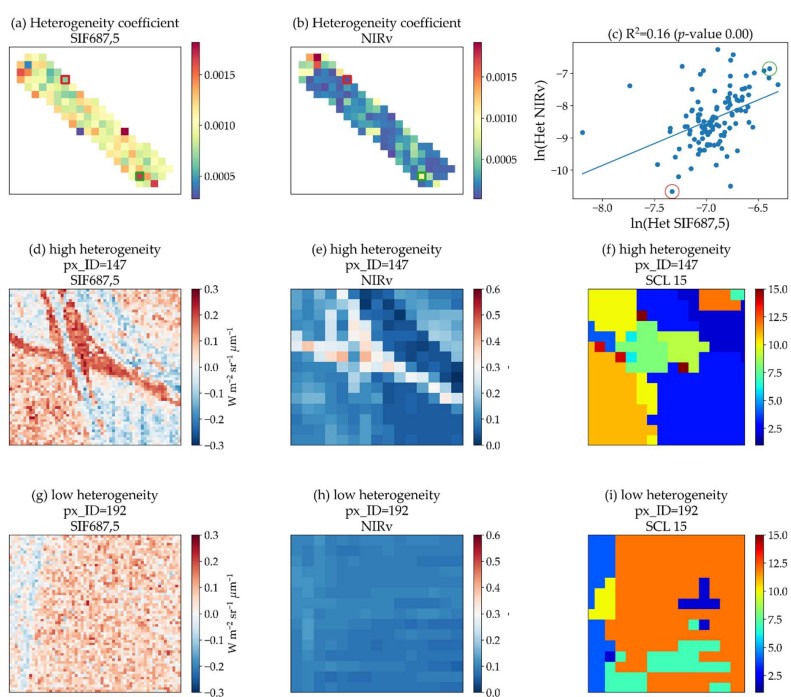

**Figure A3.** Heterogeneity maps (300 × 300 m) for the spatial heterogeneity coefficient method: (**a**) Reference SIF$_{687,5}$; (**b**) Best predictor NIRv; (**c**) Scatter plot with lowest (red circle) and highest (green circle) heterogeneity pixels highlighted; (**d**) 5 m pixel with high heterogeneity for SIF$_{687,5}$; (**e**) 20 m pixel with high heterogeneity for NIRv; (**f**) Scene classification with 15 classes for a pixel with high heterogeneity; (**g**) 5 m pixel with low heterogeneity for SIF$_{687,5}$; (**h**) 20 m pixel with low heterogeneity for NIRv; (**i**) Scene classification with 15 classes for a pixel with low heterogeneity.

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
