# Peer review of "Using Sentinel-2-Based Metrics to Characterize the Spatial Heterogeneity of FLEX Sun-Induced Chlorophyll Fluorescence on Sub-Pixel Scale"

_remotesensing, doi:10.3390/rs15194835_

Round 1

Reviewer 1 Report

Using Sentinel-2-based metrics to characterize the spatial heterogeneity of FLEX Sun-Induced chlorophyll Fluorescence at sub-pixel scale:

·        Add/Replace the name of the study area with the Keywords.

·        In the last paragraph of the Introduction, the authors should mention the weak point of former works (identification of the gaps) and describe the novelties of the current investigation to justify that the paper deserves to be published in this journal.

·        “The other goodness of fit metrics (coefficient of determination, root-mean-square error, bias) are not applicable for this task due to differences in units of SIF and predictors.”. Explain.

·        “Information entropy or class frequency ex-plains how much each land cover type contributes to the pixels.”. Explain.

·        “Notice that we decided to use fAPAR instead of fCover because the former presented slightly better results for SIF”. Explain.

·        “the number of classes that can best characterize SIF intrapixel heterogeneity, bearing in mind that more classes inherently lead to more heterogeneity, remains unclear and might be scene dependent.”. Explain.

·        Focus on the advantages/disadvantages of the proposed method concerning the obtained results.

·        What are the strategies/recommendations to reduce uncertainties in this study?

·        How can expand the results to other regions with similar/different climates?

·        At the end of the manuscript, explain the implications and future works considering the outputs of the current study.

Acceptable.

Reviewer 2 Report

This study presented a high spatial resolution downscaling method by using the S2 data as an indicator to get the fine spatial resolution SIF data. This is a well attempt to get the fine spatial resolution SIF data. I agree this method will provide some metrics in getting the fine spatial resolution SIF data.

However, the writing is confusing, the author must pay attention to describe the methods and the results.

There are some points for the authors improve their study.

Figure 2 I suggest adding a figure caption with what is a-f in this figure instead of writing the caption at the main text after this figure.

Table 1 it would be great to mention what is wavelengths of B1-B8 at S2 data at the table caption.

Table 3 is so disordered, so please revise it.

Figure 6 I don’t understand why the authors put a histogram here.

The explanation in results section should be reorganized, since the logic flow is disordered.

The English writing is okay, but the logic flow is confusing. The writing should focus on the main point on describing the method itself.

Reviewer 3 Report

Interesting article, I believe a lot of work is behind it, regarding all analyses and comparative test. I am not able to understand everything, because I do not work with spatial image sensor data and I do not use classification methods, but I really admire the work from the perspective that I work with fluorescence data, that are measured from the distance and this shows how the pixel may look like. I recommend the article for publication after considering few comments, which are related to early stages of the manuscript.

Line 29 – I do not understand what is meant by „making homogeneous pixels more reliable“. I would prefer to make introduction to the sentence like „ homogeneous signals, or low variability of signal producers….“ And then continue. I would build the sentence differently.

Line 41 – I am missing verb in this sentence

Line 58 – maybe without „having“

Around line 80 or 450, you may consider adding either of the references (https://doi.org/10.1016/j.scitotenv.2023.166386, https://doi.org/10.3390/rs15010067), which touches similar topics of variabiliy within pixels, and how it alters detected fluorescence, this time dependent on shadow occurence within pixel. But later there is bigger focus on NDVI and NIRV and its deviations, so it depends on consideration, but the topic is also touched.

Later I indeed did not find any errors in the text and the thoughts look fine.

Line 415 – „as well as“ is not appropriate in scientific literature and here it does not make sense.

Line 569 – structurE related

Round 2

Reviewer 1 Report

Acceptable!

Reviewer 2 Report

The authors addressed all of my concerns.